# Differences in Prevalence and Associated Factors of Underweight and Overweight/Obesity among Bangladeshi Adults by Gender: Analysis of a Nationally Representative Survey

**DOI:** 10.3390/ijerph191710698

**Published:** 2022-08-27

**Authors:** Rajat Das Gupta, Shams Shabab Haider, Sumaiya Zabin Eusufzai, Ehsanul Hoque Apu, Nazeeba Siddika

**Affiliations:** 1Department of Epidemiology and Biostatistics, Arnold School of Public Health, University of South Carolina, Columbia, SC 29208, USA; 2Johns Hopkins Bloomberg School of Public Health, Johns Hopkins University, 615 N Wolfe St, Baltimore, MD 21205, USA; 3Department of Biostatistics, School of Dental Sciences, University Sains Malaysia, Kota Bharu 16150, Malaysia; 4Department of Biomedical Engineering, Institute of Quantitative Health Science and Engineering, Michigan State University, East Lansing, MI 48824, USA; 5Department of Internal Medicine, Division of Hematology and Oncology, University of Michigan, Ann Arbor, MI 48109, USA; 6Department of Epidemiology and Biostatistics, College of Human Medicine, Michigan State University, East Lansing, MI 48824, USA

**Keywords:** underweight, overweight, obesity, gender, Bangladesh

## Abstract

The objective of this study was to find the differences in prevalence and associated factors of underweight and overweight/obesity among Bangladeshi adults by gender, using the nationally representative Bangladesh Demographic and Health Survey 2017–2018 data. To identify the factors associated with underweight and overweight/obesity in both genders, multilevel multivariable logistic regression was conducted. The prevalence of underweight was 19.79% and 15.49% among males and females, respectively. The prevalence of overweight/obesity was 32.67% and 45.60% among males and females, respectively. Among both genders, participants with the highest likelihood of overweight/obesity were aged 30–49 years and 50–69 years, had the highest educational attainment up to primary and secondary level, resided in a household that belonged to the middle, richer, or richest wealth quintiles, and were currently married. On the other hand, among both genders, increased educational attainment and wealth index were inversely associated with being underweight. Health promotion programs in Bangladesh should focus on these high-risk groups to address the burden of underweight and overweight/obesity.

## 1. Introduction

Underweight and overweight/obesity are two spectrums of nutritional disorders, which are associated with several morbidities and increased mortality in adults [1]. Overweight/obesity is a risk factor of several non-communicable diseases, including cardiovascular diseases, diabetes mellitus, chronic kidney disease, and osteoporosis [2]. Undernutrition also leads to insulin resistance and lower fat oxidation, which in turn increases the risk of diabetes, hypertension, dyslipidemia, and physiological impairments [3]. Underweight and overweight/obesity in women leads to complications like pre-eclampsia, eclampsia, gestational diabetes mellitus, pre-eclampsia, eclampsia, and mortality among the neonates and infants [4]. In 2016, overweight/obesity affected around 1.9 billion adults (39% of the total population), while underweight affected an additional 462 million adults (7.5% of the total population) worldwide [5].

In past decades, the nutrition status in most of the low- and middle-income countries (LMICs) improved remarkably, with decreasing prevalence of underweight. The achievement came with an unintended consequence: rising burden of overweight/obesity [6]. Still, a significant number of people around the world continued to suffer from underweight. This situation is known as the ‘double burden of malnutrition’ [7].

Bangladesh, a South Asian country, is suffering from epidemiological and nutritional transition. According to the nationally representative Bangladesh Demography and Health Survey (BDHS) 2011, the prevalence of underweight was 30.4%, overweight was 18.9% and obesity was 4.6% among adults [8]. According to BDHS 2017–2018, the prevalence of overweight/obesity exceeded the prevalence of underweight [9]. In both surveys, it was found that females were approximately twice as likely to be overweight/obese compared to males, although no such association was seen in the case of underweight [8,9]. Nevertheless, understanding the variation in the associations between deterministic factors and underweight or overweight/obesity among males and females separately is very important to designing appropriate intervention programs for vulnerable groups. This will also help the country achieve sustainable development goals [10]. However, to our knowledge, no previous studies have studied the male/female differences in the association between sociodemographic factors and underweight or overweight/obesity among Bangladeshi adults.

The objective of this study was to find the differences in prevalence and associated factors of underweight and overweight/obesity among Bangladeshi adults by gender, using a nationally representative sample.

## 2. Materials and Methods

### 2.1. Study Design

A secondary analysis of the Bangladesh Demographic and Health Survey (BDHS) 2017–2018 was carried out. The BDHS 2017–2018 was a nationally representative survey conducted in Bangladesh to update maternal, neonatal, and child health indicators. The survey also collected data on non-communicable diseases. The BDHS 2017–2018 was collected between October 2017 and October 2018. Permission to analyze the dataset for this study was taken from the DHS program in October 2021. Data analysis was conducted in October 2021. A detailed methodology of the BDHS 2017–2018, including study design, sampling, and data collection, was published previously [11]. In brief, the survey followed two-staged stratified cluster sampling. At first, a sampling frame was created based on the 2011 Population and Housing Census of Bangladesh to list all the enumeration areas (EAs). Each EA comprised an average of 120 households. In total, 675 EAs were selected (urban: 250; rural: 425). These EAs served as primary sampling units (PSUs). At the second stage, 30 households were selected randomly from each EA (total 20,250 households). From one-fourth of the selected households, all adults (aged ≥18 years) were selected for anthropometric measurement [11].

### 2.2. Outcome of Interest

Body Mass Index (BMI) was the main outcome of interest, which was calculated by dividing an individual’s body weight in kilogram (kg) divided by height squared in meter squared (m2). An Asia specific BMI cut-off was used to categorize the BMI into underweight (<18.5 kg/m^2^), normal weight (18.5 ≤ BMI < 23.0 kg/m^2^), and overweight/obese (≥23.0 kg/m^2^) [12]. Height was measured using ShorrBoard^®^ measuring board (Shorr production, Olney, Maryland, USA) in standing position. Weight was measured by lightweight, electronic SECA 878 scale (seca Deutschland, Hamburg, Germany) [11].

### 2.3. Explanatory Variables

The following covariates were considered based on the literature review: age group in years (18–29 years, 30–49 years, 50–69 years, ≥70 years); highest educational attainment (no formal education, primary, secondary, college and higher); household wealth index (poorest, poorer, middle, richer and richest); current working status (no, yes); division of residence (Barisal, Chattogram, Dhaka, Khulna, Mymensingh, Rajshahi, Rangpur and Sylhet); place of residence (urban, rural); and marital status (never married, currently married, separated/divorced/widowed) [9,13,14]. BDHS 2017–2018 collected data on selected household assets (i.e., household construction materials, water source, type of sanitation facilities, electricity use, health facilities, etc.). Principal component analysis was conducted to construct the household wealth index and was included in the dataset [11,15,16,17]. Household wealth quintiles were created by categorizing the household wealth index into five quintiles: poorest, poorer, middle, richer, richest.

### 2.4. Statistical Analysis

All the statistical analyses were conducted using STATA version 16.0 (StataCorp LLC, College Station, TX, USA). All the survey weights of the BDHS 2017–2018 were adjusted during the investigation. Initially, descriptive analyses were performed. The findings were reported in mean with standard deviation for the continuous variables. The results were reported in frequency with percentages for the categorical variables according to gender. The prevalence of underweight, overweight, and obesity was calculated, and the distribution of BMI was calculated across the covariates. Then, to find the associated factors with underweight, overweight, and obesity according to gender, multilevel logistics regression was conducted considering the hierarchical structure of the DHS data. First, unadjusted analyses were conducted and a crude odds ratio (COR) was reported. The variables which yielded a *p*-value < 0.2 were included in the final multivariable model to yield the Adjusted Odds Ratio (AOR). This cut-off of *p*-value < 0.2 was considered sufficient to control residual confounding [18]. Both COR and AOR were reported with a 95% confidence interval (CI). A *p*-value < 0.05 was considered statistically significant.

### 2.5. Ethical Consideration

The institutional review board (IRB) at ICF (IRB: FWA00000845) and the Bangladesh Medical Research Council (IRB: BMRC/NREC/2016–2019/324) approved the proposal of BDHS 2017–2018. Before data collection, informed consent was taken from the respondents. In October 2021 we obtained permission to use the dataset from the DHS program.

## 3. Results

The weighted sample of 12,458 participants was included in the final analysis. In our sample, the overall prevalence of overweight/obesity was 39.87% and the prevalence of underweight was 17.40%. The prevalence of underweight among males and females was 19.79% and 15.49%, respectively. The prevalence of overweight/obesity among males and females was 32.67% and 45.60%, respectively.

Table 1 highlights the sample characteristics according to the prevalence of different BMI categories according to gender. Among both genders, the highest burden of underweight was observed among rural residents (male: 22.0%; female: 17.3%) (among the place of residence categories), those who did not receive any formal education (male: 29.1%; female: 22.0%) (among the highest educational attainment categories), those who belonged to the poorest wealth quintile (male: 30.8%; female: 26.2%) (among all wealth quintiles), and those who were residents of Mymensingh division (male: 27.4%; female: 23.6%) (among all divisions of residence). On the other hand, with regards to age group, the highest prevalence was observed among ≥70-year-olds for males (23.3%) and among the 18–29 year olds for females (19.8%). At the same time, the burden of being underweight was higher among males who were working during the time of the survey (23.9%), compared to females who were working during the time of the survey (16.5%). Regarding marital status, the prevalence of underweight was the highest among separated/divorced/widowed males (26.5%) and single females (31.4%).

For overweight/obesity in both genders, the prevalence was highest among the 30–49 years old (male: 34.4%; female: 48.3%) (among all age groups), residents of urban areas (male: 42.2%; female: 57.1%) (among the place of residence categories), those from the richest wealth quintile (male: 56.6%; female: 68.4%) (among all wealth quintiles), and those who were currently married (male: 34.4%; female: 49.4%) (among all marital status categories). Among the highest educational attainment categories, males educated up to college and higher had the highest burden of overweight and obesity (52.8%), while for females the highest burden was observed among those who received education up to the secondary level (53.3%). The prevalence of overweight/obesity was higher among females who worked during the time of the survey (47.5%), compared to males who worked during the time of the survey (33.1%). Among the residence divisions, males of Chattogram (40.2%) and females of Khulna (51.5%) had the highest prevalence.

### 3.1. Factors Associated with Underweight

Table 2 highlights the multivariable multilevel logistics regression results regarding the factors associated with being underweight among males and females. In both genders, being underweight was associated with the highest educational attainment and household wealth status. The odds of being underweight decreased with both increasing educational attainment and wealth status. In both males and females, respondents being educated up to college and higher were 40% less likely to be underweight (male: AOR: 0.6, 95% CI: 0.4–0.8, *p* < 0.001; female: AOR: 0.6, 95% CI: 0.5–0.8, *p* < 0.01). Similarly, among both genders, respondents who belonged to the richest wealth quintiles were 50% less likely to be underweight (male: AOR: 0.5, 95% CI: 0.3–0.6, *p* < 0.001; female: AOR: 0.5, 95% CI: 0.4–0.7, *p* < 0.001).

In the case of males, current working status and division of residence were also significantly associated with undernutrition. Male respondents who were working during the time of the survey had 30% lower odds of being underweight compared to the males who were not working (AOR: 0.7, 95% CI: 0.6–0.9, *p* < 0.01). Male respondents residing in Barisal and Rangpur division were 30% less likely to be underweight than the male respondents residing in Dhaka division (Barisal: AOR: 0.7, 95% CI: 0.5–1.0, *p* < 0.05; Rangpur: AOR: 0.7, 95% CI: 0.5–1.0, *p* < 0.05).

On the other hand, age and marital status were found to be associated with being underweight in the case of females. Female respondents aged 30–49 years were 30% less likely to be underweight than female respondents aged 18–29 (AOR: 0.7, 95% CI: 0.5–0.8, *p* < 0.001). Compared to single female participants, the odds of underweight was lower among the currently married (AOR: 0.4, 95% CI: 0.3–0.6, *p* < 0.001) and separated/divorced/widowed females (AOR: 0.6, 95% CI: 0.4–0.8, *p* < 0.01).

### 3.2. Factors Associated with Overweight and Obesity

Table 3 highlights the factors associated with overweight and obesity in both genders among the adult Bangladeshi population. Except for place of residence and current employment status, all the covariates were statistically significantly associated with overweight/obesity.

Among both genders, the odds of overweight and obesity was higher among those aged 30–49 years (AOR: male: 1.4, 95% CI: 1.1–1.9, *p* < 0.01; female: 1.5, 95% CI: 1.2–1.8, *p* < 0.001) and 50–69 years (AOR: male: 1.4, 95% CI: 1.1–1.9, *p* < 0.01; female: 1.4, 95% CI: 1.1–1.7, *p* < 0.01), compared to males aged 18–29 years. Only among females was there a significant association found for respondents aged ≥70 years (AOR: 1.3, 95% CI: 1.0–1.7, *p* < 0.05).

The odds of overweight/obesity were directly associated with highest educational attainment. Among both genders, when compared to individuals receiving no formal education, the odds were significantly higher for individuals who had received education up to primary level (AOR: male: 1.4, 95% CI: 1.2–1.7, *p* < 0.01; female: 1.2, 95% CI: 1.1–1.4, *p* < 0.01) and up to secondary level (AOR: male: 1.4, 95% CI: 1.2–1.7, *p* < 0.01; female: 1.2, 95% CI: 1.1–1.4, *p* < 0.01). However, the odds were significantly higher among males educated up to college and higher levels (AOR: male: 3.5, 95% CI: 2.7–4.4, *p* < 0.001).

With an increasing wealth index, the likelihood of overweight and obesity increased in both genders. Compared to the poorest quintile, the odds were significantly higher among all wealth quintiles, except for the poorer quintile among females. Four-fold odds were observed for the highest wealth quintile (‘richest’) for both males (AOR: 4.1, 95% CI: 3.1–5.3, *p* < 0.001) and females (AOR: 4.2, 95% CI: 3.4–5.2, *p* < 0.001).

Males living in Mymensingh division had 30% lower odds of being overweight/obese than males in Dhaka division (AOR: 0.7, 95% CI: 0.5–1.0, *p* < 0.05). In Sylhet division, the odds were 20% less for females (AOR: 0.8, 95% CI: 0.6–1.0, *p* < 0.05) compared to females of Dhaka division.

Among both genders, the currently married participants had higher odds of being overweight/obese compared to single participants (AOR: male: 2.5, 95% CI: 2.0–3.0, *p* < 0.001; female: 3.5, 95% CI: 2.6–4.6, *p* < 0.001). Being separated/divorced/widowed was associated with a significant increase in the odds of overweight and obesity among females only (AOR: 2.3, 95% CI: 1.6–3.1, *p* < 0.001).

## 4. Discussion

This study aimed to identify the male-female differences in prevalence and associated factors of underweight and overweight/obesity among Bangladesh adults. We found that in most categories the prevalence of underweight was higher among males, and the prevalence of overweight/obesity was higher among females. Among both genders, participants aged 30–49 years and 50–69 years reaching the highest educational attainment of up to primary and secondary level, residing in a household that belonged to the middle, richer, or richest wealth quintiles, and being currently married, had a higher likelihood of overweight/obesity. On the other hand, among both genders, increased educational attainment and wealth index were inversely associated with being underweight.

The likelihood of being overweight/obese increases with age. Participants aged 30–49 years and 50–69 years had higher odds of overweight and obesity than participants aged 18–29. This is called the ‘inverted U-shaped pattern’ of high BMI distribution, where the prevalence is higher in the middle age groups, with decreasing prevalence in younger (18–29 years) and older (≥70 years) adults. This phenomenon was observed in Bangladesh [9] and its neighbors, including India, China, and Nepal [19,20,21]. Similarly, among both genders, the prevalence of overweight/obesity exceeded that of the underweight. Females had a higher prevalence of overweight/obesity than males. This is aligned with global statistics, where the global burden of overweight and obesity is higher among women than men [22,23]. Awareness should be raised among middle-aged adults (30–69 years) and females about the harmful effect of overweight and obesity.

The probability of being overweight/obese increased with increasing education and wealth index. Among both genders, reaching highest educational attainment up to primary and secondary level, and residing in a household that belonged to the middle, richer, or richest wealth quintiles, were significantly associated with increased risk of overweight and obesity. On the other hand, the odds of being underweight decreased with both increasing education and wealth index. In LMICs like Bangladesh, individuals with higher educational levels and higher wealth index are more likely to spend a greater amount of time with sedentary activities. Also, they have greater access to energy-dense, nutrient-poor foods [24,25]. As a result, these groups’ propensity to gain weight is higher. This is in contrast to the higher-income nations (i.e., USA, Canada), where individuals with a lower wealth index have a higher burden of overweight/obesity [26].

Being married was associated with a higher likelihood of overweight/obesity among both genders. In the case of females, being currently married and separated/divorced/widowed was inversely associated with being underweight. This is aligned with previous studies done in Bangladesh [9,27]. The propensity to eat food increases in the presence of a spouse, leading to weight gain [28]. Health promotion programs should target ever-married individuals.

Some covariates were selectively associated with males or females. Male respondents who were working during the time of the survey had 30% lower odds of being underweight than the males who were not working. This might be due to men’s weight loss due to manual labor [29]. Similarly, further exploration should be done to determine the regional differences in underweight, overweight, and obesity in both genders.

Although the prevalence of underweight has declined in Bangladesh (particularly among children and women), the burden of overweight and obesity is increasing daily [30,31]. The high burden of overweight and obesity exacerbates the nation’s struggle toward reducing the burden of non-communicable diseases. The current nutritional intervention in Bangladesh focuses mainly on the underweight rather than the overweight and obese [31]. For instance, the implementation process of mass education and awareness campaigns on physical activity is slow [32]. Implementation science and translational research should be added to overweight and obesity control programs, to understand the implementation facilitators and barriers.

This study has several notable strengths. The BDHS 2017–2018 utilized a nationally representative sample, which enabled us to generalize the findings to the target population of Bangladesh. The measurement error probability was minimal in this study, due to the utilization of validated questionnaires and calibrated instruments for data collection. The limitation of the study also warrants discussion. First, drawing a causal inference is difficult due to the study’s cross-sectional nature. Second, the data on several vital covariates related to nutritional statuses, such as diet and physical activity, were not collected by BDHS 2017–2018. As a result, we could not include those variables in our final model. 

## 5. Conclusions

This study found a high prevalence of overweight/obesity among both Bangladeshi males and females. Approximately every one-in-two females, and every one-in-three males, were either overweight or obese. Although the prevalence of underweight was lower than that of overweight/obesity, it is still quite high among both genders (male: 19.79%; female: 15.49%). Among both genders, participants aged 30–49 years and 50–69 years, reaching the highest educational attainment of up to primary and secondary level, residing in a middle, richer, or richest wealth quintile household, and being currently married, had the higher likelihood of overweight/obesity. Equitable and cost-effective public health nutrition programs in Bangladesh should focus on these high-risk groups to address the high burden of overweight/obesity. On the other hand, individuals from the poorest households and no formal education should be targeted by the public health promotion programs addressing undernutrition.

## Figures and Tables

**Table 1 ijerph-19-10698-t001:** Sample characteristics according to prevalence of different categories body mass index according to gender, BDHS 2017–2018.

Variables	Male (n = 5528)						Female (n = 6930)					
	Underweight		Normal Weight		Overweight/Obesity		Underweight		Normal Weight		Overweight/Obesity	
	n	%	n	%	n	%	n	%	n	%	n	%
**Age (in Years)**												
18–29	106	22.3	260	54.6	110	23.1	141	19.8	313	43.9	260	36.4
30–49	433	17.6	1185	48.0	849	34.4	430	13.2	1249	38.5	1568	48.3
50–69	431	21.0	953	46.4	670	32.6	401	17.0	898	38.1	1059	44.9
70+	123	23.3	229	43.3	177	33.5	102	16.6	237	38.6	274	44.7
**Place of Residence**												
Urban	222	14.2	682	43.6	659	42.2	195	10.6	599	32.4	1054	57.1
Rural	871	22.0	1947	49.1	1147	28.9	879	17.3	2097	41.3	2106	41.4
**Highest Educational Attainment**												
No Formal Schooling	372	29.1	671	52.5	235	18.4	428	22.0	835	42.9	684	35.1
Primary	389	22.5	884	51.1	457	26.4	301	15.0	799	39.9	903	45.1
Secondary	226	15.1	697	46.6	574	38.3	233	11.0	754	35.7	1127	53.3
College and Higher	107	10.4	376	36.8	541	52.8	111	12.8	308	35.6	446	51.6
**Household Wealth Index**												
Poorest	312	30.8	545	53.8	156	15.4	355	26.2	609	44.9	393	29.0
Poorer	278	25.8	575	53.3	226	20.9	266	19.7	633	46.9	451	33.4
Middle	234	20.3	587	50.9	332	28.8	196	14.0	578	41.3	625	44.7
Richer	174	15.6	510	45.7	433	38.8	145	10.8	519	38.7	677	50.5
Richest	96	8.2	411	35.2	660	56.6	112	7.5	357	24.1	1014	68.4
**Current Working Status**												
No	187	23.9	361	46.1	235	30.1	591	14.7	1515	37.8	1906	47.5
Yes	907	19.1	2268	47.8	1571	33.1	482	16.5	1181	40.5	1255	43.0
**Division of Residence**												
Barisal	58	18.9	156	50.9	92	30.2	58	15.1	146	38.0	180	46.9
Chattogram	148	17.1	369	42.6	348	40.2	147	11.5	491	38.4	640	50.1
Dhaka	259	19.2	583	43.1	510	37.7	217	13.3	574	35.3	837	51.4
Khulna	124	17.8	325	46.5	250	35.7	107	12.8	297	35.7	428	51.5
Mymensingh	125	27.4	231	50.7	100	21.9	133	23.6	237	42.1	194	34.4
Rajshahi	158	20.0	412	52.0	223	28.1	163	16.6	405	41.3	413	42.1
Rangpur	135	19.6	362	52.6	192	27.8	153	18.7	359	43.8	308	37.6
Sylhet	86	23.5	190	51.7	92	24.9	96	21.7	187	42.2	160	36.2
**Marital Status**												
Never Married	207	22.9	458	50.7	239	26.4	111	31.4	164	46.2	79	22.4
Currently Married	853	19.0	2097	46.6	1547	34.4	725	13.1	2072	37.5	2725	49.4
Separated/Divorced/Widowed	34	26.5	73	57.2	21	16.4	238	22.5	461	43.7	356	33.8

BDHS: Bangladesh Demographic and Health Survey.

**Table 2 ijerph-19-10698-t002:** Factors associated with underweight among both male and female, BDHS 2017–2018.

Variables	Male		Female	
	COR	AOR	COR	AOR
**Age (in Years)**				
18–29	**Ref**	**Ref**	**Ref**	**Ref**
30–49	0.9 (0.7–1.1)	0.9 (0.7–1.1)	0.7 ** (0.6–0.9)	0.7 *** (0.5–0.8)
50–69	1.1 (0.8–1.4)	1.1 (0.9–1.4)	0.9 (0.7–1.1)	0.9 (0.7–1.1)
70+	1.3+ (0.9–1.7)	1.2 (0.8–1.6)	0.9 (0.7–1.2)	0.8 (0.6–1.2)
**Place of Residence**				
Urban	**Ref**	**Ref**	**Ref**	**Ref**
Rural	1.3 *** (1.1–1.6)	1.0 (0.9–1.3)	1.3 *** (1.1–1.6)	1.1 (0.9–1.3)
**Highest Educational Attainment**				
No Formal Schooling	**Ref**	**Ref**	**Ref**	**Ref**
Primary	0.8 * (0.7–1.0)	0.9 (0.7–1.1)	0.7 (0.6–0.9)	0.8 * (0.7–1.0)
Secondary	0.6 *** (0.5–0.7)	0.7 * (0.6–0.9)	0.6 (0.5–0.7)	0.7 ** (0.6–0.8)
College and Higher	0.5 *** (0.4–0.6)	0.6 *** (0.4–0.8)	0.6 (0.5–0.8)	0.6 ** (0.5–0.8)
**Household Wealth Index**				
Poorest	**Ref**	**Ref**	**Ref**	**Ref**
Poorer	0.9 (0.7–1.1)	0.9 (0.7–1.1)	0.7 ** (0.6–0.9)	0.7 ** (0.6–0.9)
Middle	0.7 ** (0.6–0.9)	0.7 ** (0.6–0.9)	0.6 *** (0.5–0.8)	0.6 *** (0.5–0.8)
Richer	0.6 *** (0.5–0.8)	0.7 ** (0.5–0.8)	0.4 *** (0.3–0.5)	0.5 *** (0.3–0.6)
Richest	0.4 *** (0.3–0.6)	0.5 *** (0.3–0.6)	0.5 *** (0.4–0.6)	0.5 *** (0.4–0.7)
**Current Working Status**				
No	**Ref**	**Ref**	**Ref**	Not included in the final sample
Yes	0.8+ (0.7–1.0)	0.7 ** (0.6–0.9)	1.0 (0.9–1.2)	
**Division of Residence**				
Dhaka	**Ref**	**Ref**	**Ref**	**Ref**
Barisal	0.9 (0.6–1.2)	0.7 * (0.5–1.0)	1.1 (0.8–1.5)	0.9 (0.7–1.3)
Chattogram	1.0 (0.7–1.3)	0.9 (0.6–1.2)	0.8 (0.6–1.1)	0.8 (0.6–1.1)
Khulna	0.9 (0.7–1.2)	0.8 (0.6–1.1)	1.0 (0.7–1.3)	1.0 (0.7–1.3)
Mymensingh	1.3+ (0.9–1.7)	1.0 (0.7–1.3)	1.5 ** (1.1–2.0)	1.2 (0.9–1.6)
Rajshahi	0.9 (0.7–1.2)	0.8 (0.6–1.1)	1.1 (0.8–1.5)	1.0 (0.7–1.4)
Rangpur	0.9 (0.7–1.2)	0.7 * (0.5–1.0)	1.2 (0.9–1.6)	0.9 (0.7–1.2)
Sylhet	1.1 (0.8–1.5)	1.0 (0.7–1.3)	1.3 * (1.0–1.8)	1.1 (0.8–1.5)
**Marital Status**				
Never Married	**Ref**	Not included in the final sample	**Ref**	**Ref**
Currently Married	1.0 (0.8–1.2)		0.6 *** (0.4–0.7)	0.4 *** (0.3–0.6)
Separated/Divorced/Widowed	1.2 (0.7–1.8)		0.9 (0.6–1.1)	0.6 ** (0.4–0.8)

AOR: Adjusted Odds Ratio; BDHS: Bangladesh Demographic and Health Survey; COR: Crude Odds Ratio; CI: Confidence Interval. *p*-value: + < 0.2, * < 0.05, ** < 0.01, *** < 0.001.

**Table 3 ijerph-19-10698-t003:** Factors associated with overweight/obesity among both male and female, BDHS 2017–2018.

Variables	Male		Female	
	COR	AOR	COR	AOR
**Age (in Years)**				
18–29	**Ref**	**Ref**	**Ref**	**Ref**
30–49	1.7 *** (1.3–2.1)	1.4 ** (1.1–1.9)	1.6 *** (1.3–2.0)	1.5 *** (1.2–1.8)
50–69	1.7 *** (1.3–2.1)	1.4 ** (1.1–1.9)	1.5 *** (1.2–1.9)	1.4 ** (1.1–1.7)
70+	1.5 * (1.1–2.1)	1.3 (0.9–1.8)	1.5 ** (1.1–1.9)	1.3 * (1.0–1.7)
**Place of Residence**				
Urban	**Ref**	**Ref**	**Ref**	**Ref**
Rural	0.6 *** (0.5–0.6)	0.9 (0.8–1.1)	0.6 *** (0.5–0.6)	0.9 (0.8–1.0)
**Highest Educational Attainment**				
No Formal Schooling	**Ref**	**Ref**	**Ref**	**Ref**
Primary	1.5 *** (1.2–1.8)	1.4 ** (1.2–1.7)	1.4 *** (1.2–1.7)	1.2 ** (1.1–1.4)
Secondary	2.4 *** (2.0–3.0)	1.9 *** (1.6–2.4)	1.9 *** (1.6–2.2)	1.4 *** (1.2–1.6)
College and Higher	4.3 *** (3.5–5.2)	3.5 *** (2.7–4.4)	1.8 *** (1.5–2.2)	1.2 (1.0–1.5)
**Household Wealth Index**				
Poorest	**Ref**	**Ref**	**Ref**	**Ref**
Poorer	1.5 ** (1.2–1.9)	1.4 * (1.1–1.8)	1.1 (0.9–1.3)	1.0 (0.9–1.2)
Middle	2.1 *** (1.6–2.6)	1.8 *** (1.4–2.3)	1.7 *** (1.4–2.0)	1.6 *** (1.3–1.9)
Richer	3.1 *** (2.5–3.9)	2.4 *** (1.8–3.1)	2.0 *** (1.6–2.3)	1.8 *** (1.5–2.2)
Richest	6.4 *** (5.1–8.1)	4.1 *** (3.1–5.3)	4.4 *** (3.7–5.3)	4.2 *** (3.4–5.2)
**Current Working Status**				
No	**Ref**	**Ref**	**Ref**	**Ref**
Yes	1.2 + (1.0–1.4)	1.2 (1.0–1.5)	0.9 * (0.8–1.0)	1.0 (0.9–1.2)
**Division of Residence**				
Dhaka	**Ref**	**Ref**	**Ref**	**Ref**
Barisal	0.7 * (0.5–1.0)	1.1 (0.8–1.5)	0.8 (0.7–1.1)	1.3 (1.0–1.6)
Chattogram	1.0 (0.8–1.4)	1.2 (0.9–1.6)	0.9 (0.7–1.1)	1.0 (0.8–1.3)
Khulna	0.9 (0.7–1.2)	1.1 (0.8–1.4)	1.0 (0.8–1.3)	1.2 (1.0–1.5)
Mymensingh	0.5 *** (0.4–0.7)	0.7 * (0.5–1.0)	0.5 *** (0.4–0.7)	0.8 (0.6–1.0)
Rajshahi	0.6 ** (0.5–0.8)	0.8 (0.6–1.1)	0.7 ** (0.5–0.9)	1.0 (0.8–1.3)
Rangpur	0.7 ** (0.5–0.9)	1.0 (0.8–1.3)	0.6 *** (0.5–0.8)	0.9 (0.7–1.2)
Sylhet	0.6 *** (0.4–0.8)	0.8 (0.6–1.1)	0.6 *** (0.4–0.7)	0.8 * (0.6–1.0)
**Marital Status**				
Never Married	**Ref**	**Ref**	**Ref**	**Ref**
Currently Married	1.7 *** (1.4–2.0)	2.5 *** (2.0–3.0)	3.0 *** (2.3–3.9)	3.5 *** (2.6–4.6)
Separated/Divorced/Widowed	0.7 (0.4–1.1)	1.3 (0.8–2.2)	1.7 *** (1.3–2.3)	2.3 *** (1.6–3.1)

AOR: Adjusted Odds Ratio; BDHS: Bangladesh Demographic and Health Survey; COR: Crude Odds Ratio; CI: Confidence Interval. *p*-value: + < 0.2, * < 0.05, ** < 0.01, *** < 0.001.

## Data Availability

The deidentified data of BDHS 2017–2018 is available online: https://dhsprogram.com/data/dataset/Bangladesh_Standard-DHS_2017.cfm?flag=0 (accessed on 10 October 2021). following proper procedure.

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
