# Peer review of "Differences in Prevalence and Associated Factors of Underweight and Overweight/Obesity among Bangladeshi Adults by Gender: Analysis of a Nationally Representative Survey"

_ijerph, 2022, doi:10.3390/ijerph191710698_

Round 1
Reviewer 1 Report
Dear authors!
The aim of this study was to find out the differences in prevalence and associated factors of underweight and overweight/obesity among Bangladeshi adults by gender using the nationally representative Bangladesh Demographic and Health Survey 2017-18 data using the multilevel multivariable logistic regression. The Permission to analyze the dataset for this study was taken from the DHS program.
Self-citation is 12.5%.
The article corresponds to the profile of the journal. The presentation style is clear and consistent.
The "References" section contains 32 publications for the period from 1992 to 2022, of which 12 publications (37.5%) were published in the last 5 years (2018-2022). Links to all publications from the "References" section are presented in the article. The conclusions are justified and follow logically from the results. It is recommended for publication.
Minor comments, which do not reduce the high rating of the work:
1. At the beginning of the article there is a section “0. How to Use This Template” - as far as I understand, this section was included in the text of the article by mistake.
2. Could you comment on the following:
Lines 78-79 (2.1. Study design): «Permission to analyze the dataset for this study was taken from the DHS program in October 2021»
Line 127 (2.5. Ethical consideration): «In April 2022, we obtained permission to use the dataset from the DHS program. »
There are two different permissions? Or in case it’s one permission, why there are different dates?
It is recommended for publication.
Author Response
Reviewer 1:
Dear authors!
The aim of this study was to find out the differences in prevalence and associated factors of underweight and overweight/obesity among Bangladeshi adults by gender using the nationally representative Bangladesh Demographic and Health Survey 2017-18 data using the multilevel multivariable logistic regression. The Permission to analyze the dataset for this study was taken from the DHS program.
Self-citation is 12.5%.
The article corresponds to the profile of the journal. The presentation style is clear and consistent.
The "References" section contains 32 publications for the period from 1992 to 2022, of which 12 publications (37.5%) were published in the last 5 years (2018-2022). Links to all publications from the "References" section are presented in the article. The conclusions are justified and follow logically from the results. It is recommended for publication.
Minor comments, which do not reduce the high rating of the work:
- At the beginning of the article there is a section “0. How to Use This Template” - as far as I understand, this section was included in the text of the article by mistake.
Comment: Thank you very much for this important observation. We have deleted that section in the revised manuscript.
- Could you comment on the following:
Lines 78-79 (2.1. Study design): «Permission to analyze the dataset for this study was taken from the DHS program in October 2021»
Line 127 (2.5. Ethical consideration): «In April 2022, we obtained permission to use the dataset from the DHS program. »
There are two different permissions? Or in case it’s one permission, why there are different dates?
It is recommended for publication.
Comment: Thank you for this very important comment. We actually took permission to analyze the dataset in October 2021. We have revised the section in the ethical consideration subsection as following: “In October 2021, we obtained permission to use the dataset from the DHS program.”
Reviewer 2 Report
I find this manuscript very interesting especially for comparing the prevalence of underweight and overweight/obesity in eastern cultures such as Bangladesh with western cultures such as the USA.
I suggest minor changes to improve readers' understanding of the manuscript.
row 47 and 48 (Introduction) It would be good to have number of underweight adults presented also in percentage.
row 100 (Explantory variables) I believe that the criteria by which a certain respondent belongs to a certain category of the household wealth index (poorest, poorest, middle, richest and richest) should be better explained.
row 137-138 (Results) The authors compare the prevalence of underweight and overweight/obesity between men and women depending on the level of education.
I believe that this result is influenced by the ratio of the percentage of men to the percentage of women in the highly educated population.
If this percentage is nearly equal in Bangladesh then a reference proving that should be inserted in the manuscript.
Author Response
REVIEWER 2:
I find this manuscript very interesting especially for comparing the prevalence of underweight and overweight/obesity in eastern cultures such as Bangladesh with western cultures such as the USA.
I suggest minor changes to improve readers' understanding of the manuscript.
row 47 and 48 (Introduction) It would be good to have number of underweight adults presented also in percentage.
Comment: Thank you for this comment. We have presented the number of underweight adults in percentages. We have revised accordingly: “In 2016, overweight/obesity affected around 1.9 billion adults (39% of the total popula-tion), while underweight affected an additional 462 million adults (7.5% of the total population), worldwide [5].”
row 100 (Explantory variables) I believe that the criteria by which a certain respondent belongs to a certain category of the household wealth index (poorest, poorest, middle, richest and richest) should be better explained.
Comment: Thanks! We have revised the statement as following: “BDHS 2017-18 collected data on selected household assets (i.e., household construction materials, water source, type of sanitation facilities, electricity use, health facilities, etc.). Principal component analysis was conducted to construct the household wealth index and was included in the dataset [11,15–17]. Household wealth quintiles were created by categorizing the household wealth index into five quintiles: poorest, poorer, middle, richer, richest.”
row 137-138 (Results) The authors compare the prevalence of underweight and overweight/obesity between men and women depending on the level of education.
I believe that this result is influenced by the ratio of the percentage of men to the percentage of women in the highly educated population.
If this percentage is nearly equal in Bangladesh then a reference proving that should be inserted in the manuscript.
Comment: Thank you for this comment.
The distribution of the highest educational attainment in the sample according to gender is shown in the following table:
|
Highest Educational Attainment |
Male (%) |
Female (%) |
Total (%) |
|
No Formal Schooling |
23.1 |
28.1 |
25.9 |
|
Primary |
31.3 |
28.9 |
30.0 |
|
Secondary |
27.1 |
30.5 |
29.0 |
|
College and Higher |
18.5 |
12.5 |
15.2 |
This percentage differs from the Education household survey (EHS-2014) conducted by Bangladesh Bureau of Statistics. As BDHS 2017-18 utilized a nationally representative sample, we can assume that this finding reflects the latest estimate in the context of Bangladesh.